# Outlook on Magnetohydrodynamical Turbulence and Its Astrophysical Implications

Elena Popova [1] and Alexandre Lazarian [1,2,*]

1  Centro de Investigación en Astronomía, Universidad Bernardo O'Higgins, Santiago 8370854, Chile
2  Department of Astronomy, University of Wisconsin–Madison, Madison, WI 53706, USA
*  Correspondence: alazarian@facstaff.wisc.edu

**Abstract:** Magnetohydrodynamical (MHD) turbulence is ubiquitous in magnetized astrophysical plasmas, and it radically changes a great variety of astrophysical processes. In this review, we introduce the concept of MHD turbulence and explain the origin of its scaling. We consider the implications of MHD turbulence for various problems: dynamo in different types of stars, flare activity, solar and stellar wind from different stars, the propagation of cosmic rays, and star formation. We also discuss how the properties of MHD turbulence provide a new means of tracing magnetic fields in interstellar and intracluster media.

**Keywords:** MHD turbulence; dynamo; star; magnetic field; flare; star formation; interstellar medium

## 1. Introduction

Turbulence is ubiquitous in astrophysics. The evidence of turbulence includes a Kolmogorov spectrum of electron density fluctuations [1,2], through numerous measurements of solar wind fluctuations [3] and the non-thermal broadening of spectral lines, as well as measures obtained by other techniques; see [4]. This is due to astrophysical plasmas having very large Reynolds numbers. Plasma flows at these high Reynolds numbers are subject to numerous linear and finite-amplitude instabilities that induce turbulence.

Turbulence can be driven by an external energy source, such as supernova explosions in the ISM [5,6], merger events and active galactic nuclei outflows in the intercluster medium (ICM) [7–9], and baroclinic forcing behind shock waves in interstellar clouds. Turbulence can also be driven by a rich array of instabilities, such as magneto-rotational instability (MRI) in accretion disks [10,11], kink instability of twisted flux tubes in the solar corona [12,13], etc.

The properties of the media change dramatically in the presence of turbulence. In particular, transport processes are changed by turbulence.

Astrophysical turbulence is magnetized and therefore the turbulence in the presence of a magnetic field is the most important for astrophysical applications. In this review, we discuss the basic properties of MHD turbulence in Section 2, its relation to dynamo in Section 3, and its connection to magnetic reconnection and reconnection diffusion in Section 4. We discuss the turbulence in spiral galaxies and its effects on star formation in Section 5. Turbulence effects on cosmic ray physics are considered in Section 6, while a new means of magnetic field study that employs MHD turbulence properties, i.e., the gradient technique (GT), is discussed briefly in Section 7.

## 2. Basics of MHD Turbulence Theory

### 2.1. General Considerations

The modern understanding of MHD turbulence theory is described in the monograph by Beresnyak and Lazarian [14]. Below, we provide a brief sketch of a few fundamental ideas at the theory's foundations.

First of all, MHD turbulence is anisotropic. This property has been known for a while (see [15]) but was associated with scale-independent anisotropy that was measured in numerical simulations. In [16], the concept of scale-dependent anisotropy scaling was introduced. The compressible MHD turbulence is based on the notion of the superposition of three cascades of fundamental modes, i.e., Alfvén, slow and fast. We use term "mode" rather than "wave" as, in strong turbulence, the Alfvén turbulent perturbations undergo nonlinear damping/cascading over one period. This is definitely not wave behavior.

The Alfvén modes determine turbulence anisotropy. They enslave the cascades of slow modes and impose their anisotropy on the slow modes [16–19]. In non-relativistic turbulence, fast modes follow their own cascades that depend marginally on the cascades of Alfvén and slow modes [18]. This cascade is similar to the acoustic one in a high $\beta$ ($\beta$ is the ratio of the plasma to magnetic pressure) medium ([16]). In this regime, the fast modes are mostly compressions of media propagating with sound velocity $c_s$. In [18], it was demonstrated that in the low-$\beta$ turbulence, the fast modes form a cascade similar to the acoustic one, even though the fluctuations are magnetic compressions propagating with Alfvén velocity $V_A$. In fact, numerical simulations in [18,19] indicate that the cascade of the fast modes is very similar to the acoustic cascade for all $\beta$. Figure 1 illustrates the spectra MHD modes. We discuss their properties below.

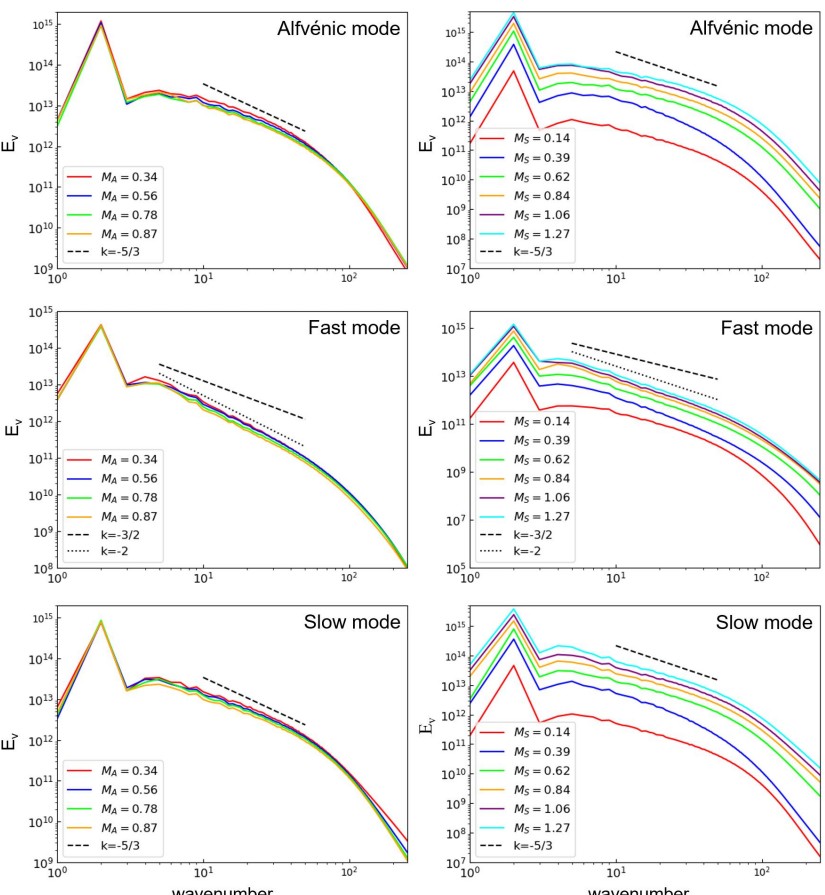

**Figure 1.** **Left:** The power spectrum of velocity fluctuations for Alfvénic (top), fast (central) and slow (bottom) modes. The sonic Mach number $M_S \approx 0.6$. **Right:** The spectra of 3 modes for $M_A \approx 0.5$. From [20].

## 2.2. Alfvén and Slow Modes

The properties of MHD turbulence change with the Alfvén Mach number $M_A = V_L/V_A$, where $V_L$ and $V_A$ are, respectively, the turbulent injection and Alfvén velocity. For $M_A \ll 1$, magnetic fields are weakly perturbed, while magnetic fields are highly chaotic for $M_A \gg 1$.

The original version in [16] was formulated by [16] in Fourier space in the system of reference of the mean magnetic field. In fact, the formulated scalings *are not valid in the system of the mean magnetic field*. The correct choice of the system of reference follows from the theory of turbulent reconnection [21]. Turbulent reconnection is part and parcel of MHD turbulence (see [22]). Ref. [21]'s theory demonstrates that in trans-Alvenic turbulence, the reconnection of eddies is equal to the eddy turnover time. Therefore, Alfvénic turbulence can be treated as the collection of eddies whose axes of rotation are aligned with the magnetic field in their vicinity. Turbulent reconnection enables the fluid motions that induce mixing perpendicular to the *local* magnetic field direction.

It is easy to see that the eddy's perpendicular magnetic field in the presence of turbulent reconnection experiences minimal resistance. Indeed, the magnetic field bending by such eddies is minimal. Thus, the energy of turbulent driving is being channeled through such eddies.

An eddy with scale $l_\perp$ perpendicular to the local magnetic field induces an Alfvénic perturbation of scale $l_\parallel$ that propagates along the magnetic field with speed $V_A$. As an eddy induces this perturbation with the turnover time $l_\perp/v_l$, this should be equal to the timescale of the corresponding Alfvénic perturbation $l_\parallel/V_A$ induced by the eddy:

$$\frac{l_\parallel}{V_A} \approx \frac{l_\perp}{v_l}. \tag{1}$$

This relation constitutes the modern understanding of the critical balance for Alfvénic turbulence.

As a result, [16]'s original theory must be augmented by the concept of a *local system of reference*. The vital significance of this system of reference for describing MHD turbulence is confirmed by numerical simulations [23–25]. One should keep in mind that the direction of the local magnetic field at a given region can differ significantly from the global mean magnetic field direction. The latter results from large-scale averaging, and it does not generally coincide with the realization of the magnetic field at a given point. Disregarding the difference between the local and global systems of reference is the most common mistake in the literature dealing with MHD turbulence. The scale-dependent anisotropy laws formulated in [16] are valid only in the local reference system.

Observational studies of the average magnetic field along the line of sight make it impossible to define the 3D local reference frame. Thus, the observationally measured anisotropy is not expected to be scale-dependent. In the reference system related to the mean field, i.e., the global system of reference, the largest eddies dominate the measured turbulence anisotropy (see [25]).

While the local system of reference is not easy to define in measuring magnetic fields from observations, this is a natural system of reference for astrophysical processes. For instance, the local direction of a magnetic field in the solar vicinity is very different from the average direction of a magnetic field in the Milky way. Cosmic ray propagation in the solar system neighborhood is determined by the local magnetic field rather than the Milky Way's averaged one.

As mentioned earlier, turbulent reconnection allows the turbulent cascade in the direction perpendicular to the local magnetic field. This cascade is not affected by the back-reaction of the magnetic field. Thus, the Alfvénic cascade is Kolmogorov-like. For trans-Alfvénic turbulence with $V_L = V_A$, this entails

$$v_l \approx V_A \left(\frac{l_\perp}{L}\right)^{1/3}, \tag{2}$$

where $v_l$ is the turbulent velocity corresponding to the perpendicular eddy $l_\perp$ scale.

In the *local system of reference*, combining Equations (1) and (2), it is easy obtain the scale-dependent anisotropy of trans-Alfvénic turbulence:

$$l_\parallel \approx L \left( \frac{l_\perp}{L} \right)^{2/3}. \tag{3}$$

This testifies that smaller eddies are more elongated along the local magnetic field.

If turbulence is injected with $M_A > 1$, the magnetic field is weak at injection scale $L$. Therefore, the super-Alfvénic turbulence at large scales evolves along an isotropic Kolmogorov energy spectrum. However, as turbulent velocity decreases at smaller scales, i.e., $v_l^2 \sim V_L(l/L)^{2/3}$, the effect of the magnetic field becomes important. At the scale [26]

$$l_A = L M_A^{-3}, \tag{4}$$

$v_l$ becomes equal to $V_A$, and the turbulence transfers to the MHD regime. In fact, at scales smaller than $l_A$, turbulence can be described by trans-Alfvénic scaling, provided that $L$ in Equations (2) and (3) is replaced by $l_A$.

The analysis of the literature shows that researchers frequently miss that the [16] scalings are not valid for sub-Alfvénic turbulence with $M_A < 1$. It was demonstrated in [21] that in the vicinity of the injection scale, $L$, the sub-Alfvénic turbulence evolves along a different type of cascade. This regime is termed the *weak* regime of Alfvénic turbulence. In this regime, the parallel scale of wave packets remains equal to the injection scale, i.e., $l_\parallel = L = const$, and the Alfvén perturbations interact multiple times to be cascaded.

For weak turbulence, the scaling obtained in [21] for the weak turbulence (weak turbulence is weak in terms of the nonlinear interactions) under the assumption of the isotropic turbulence driving at $L$ is

$$v_l \approx V_L \left( \frac{l_\perp}{L} \right)^{1/2}, \tag{5}$$

which also corresponds to the subsequent detailed analytical study in [27].

An important feature of the weak Alfvénic cascade is that, with the decrease in $l_\perp$, the intensity of interactions of Alfvénic perturbations increases. This is counterintuitive as, with the decrease in $l_\perp$, the turbulence decreases in its amplitude. Therefore, as shown in [21], at scale

$$l_{\text{tran}} \approx L M_A^2, \tag{6}$$

where $M_A < 1$, and the turbulence enters into the strong turbulence regime. For the sub-Alfvénic MHD turbulence at $l < l_{\text{tran}}$,

$$v_l \approx V_L \left( \frac{l_\perp}{L} \right)^{1/3} M_A^{1/3}, \tag{7}$$

and

$$l_\parallel \approx L \left( \frac{l_\perp}{L} \right)^{2/3} M_A^{-4/3}. \tag{8}$$

It is important to note that the relations above that were derived in [21] differ from Equations (2) and (3) for trans-Alfvénic turbulence, i.e., for $M_A = 1$, by the additional dependence on the Alfvén Mach number $M_A$.

### 2.3. Fast Modes

The numerical decomposition of MHD turbulence into modes has demonstrated that the interaction between fast modes, on the one hand, and slow, and Alfvén, on the other hand, is relatively weak for a low Alfvén Mach number $M_A = V_l/V_A$ for non-relativistic MHD turbulence [18]. The cascade of the fast modes, therefore, can be assumed independent of the cascades of the Alfvén and slow modes. This cascade is similar to the

acoustic one in a high-$\beta$ medium (we remind the reader that $\beta$ is the ratio of the plasma to magnetic pressure) [16]. This is because, in this regime, the fast modes are mostly compressions of plasmas that propagate with sound velocity $c_s$. It was also shown by [18] that in the opposite limiting case of low-$\beta$ plasmas, the fast modes are expected to form a cascade similar to the acoustic one, even though the fluctuations are compressions of the magnetic field that propagate with velocity $\sim V_A$. The numerical simulations in [18,19] support the idea that the cascade of the fast modes is very similar to the acoustic cascade for all $\beta$.

For fast modes in a high-$\beta$ medium, the perturbations are similar to sonic waves. Thus, their amplitude increases with the sonic Mach number $M_s$. For fast modes in a low-$\beta$ medium, the increase in amplitude corresponds to the increase in $M_A$. The numerical results in [19] are consistent with $E_f \sim k^{-3/2}$, while those in [28] are better fitted by $E_f \sim k^{-2}$.

The difference can be accounted for by appealing to the analogy with the acoustic turbulent cascade, but it is not a settled issue. For instance, new simulations shown in Figure 1 suggest that the spectrum of $k^{-2}$ for subsonic turbulence corresponds to $M_s = 0.6$. Whatever their exact spectral index, the fast modes' fluctuations are isotropic.

Figure 2 illustrates the anisotropies of three fundamental modes of MHD turbulence, i.e., Alfvén, slow and fast. The contours of iso-correlation are shown.

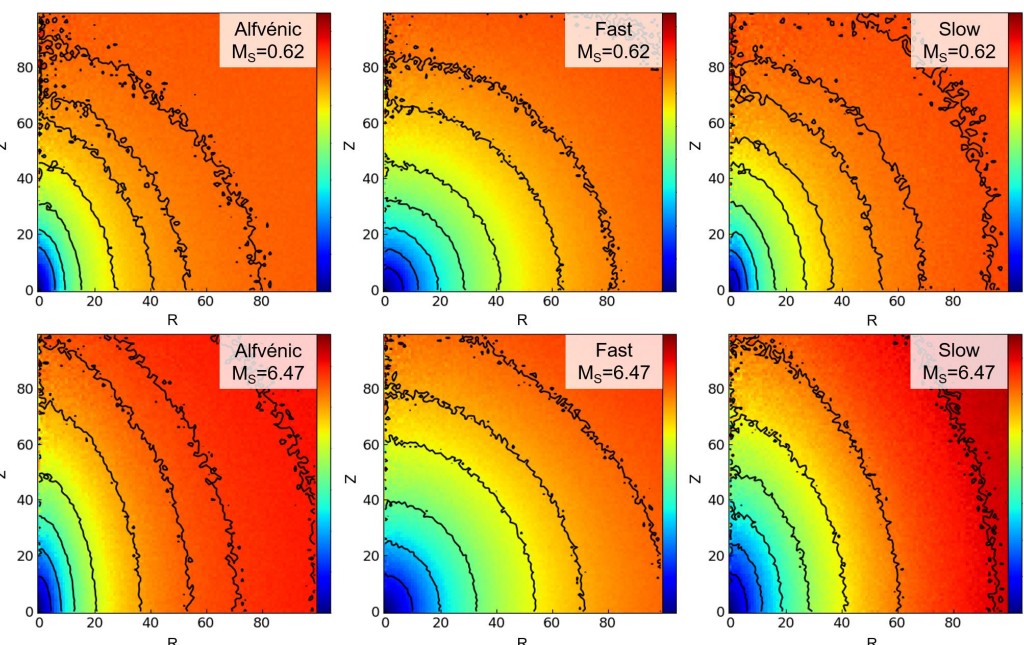

**Figure 2.** Iso-contours of equal correlation for structure functions measured in the local reference frame. The coordinate $Z$ is measured parallel to the local magnetic field, while $R$ is measured perpendicular to the local magnetic field. The turbulence corresponds to $M_A \approx 0.5$. From [20].

## 3. MHD Turbulence and Dynamo

MHD turbulence plays a key role in the turbulent dynamo. The dynamo process is a mechanism for generating a magnetic field in celestial bodies—in particular, in the Sun and stars [29,30]. The problem of explaining the occurrence of a magnetic field in celestial bodies began with the discovery of the terrestrial and solar magnetic fields.

The first theoretical work on what constitutes the Earth's magnetic field, i.e., what is the magnitude and direction of its intensity at each point on the Earth's surface, belongs to the German mathematician Carl Gauss. In 1834, he gave a mathematical expression for the components of tension as a function of coordinates—the latitude and longitude of the observation site. Using this expression, it is possible to find, for each point on the Earth's

surface, the values of any of the components called the elements of the Earth's magnetism. This and other works of Gauss became the foundation on which the modern science of terrestrial magnetism is built [31]. In particular, in 1839, he proved that the main part of the magnetic field comes out of the Earth, and the cause of small, short deviations in its values must be sought in the external environment [31].

The discovery of the solar magnetic field is associated with the observation of sunspots on its surface, which began to be conducted a very long time ago.

The first reports of sunspots date back to 800 BC in China; sunspots are mentioned in the writings of Theophrastus of Athens (4th century BC), and the oldest known drawing of sunspots was created on 8 December 1128, by John of Worcester (published in The Chronicle of John of Worcester). In 1610, astronomers began using a telescope to observe the Sun. Initial research focused on the nature of the spots and their behavior. Despite the fact that the physical nature of the spots remained unclear until the 20th century, observations continued. In the 15th and 16th centuries, research was hindered by their small number, which is now regarded as a prolonged period of low SA, called the Maunder minimum. By the 19th century, there was already a sufficiently long series of observations of the number of sunspots to determine periodic cycles in the activity of the Sun. In 1845, Professors D. Henry and S. Alexander of Princeton University observed the Sun with a thermometer. They determined that the spots emitted less energy than the surrounding areas of the Sun. Later, above-average radiation was determined in areas of the so-called solar plumes [32].

For the first time, the magnetic field of the Sun was discovered and reliably measured in 1908 by J. Hale in only one of the spots [33]. Then, the field strength was found to be 2 kilogauss, which is 2–4 thousand times greater than the Earth's magnetic field (but almost 10 times less than the field of a modern magnetic resonance imaging apparatus, around 50 times less than the strongest fields created by man and billions of times smaller than the fields of some neutron stars).

Now, the observation of sunspots and the study of their magnetic fields is one of the daily tasks of modern heliophysics [34,35].

Today, different agencies perform monitoring of solar activity—for example, the Solar Dynamics Observatory (SDO) (https://sdo.gsfc.nasa.gov/, aceessed on 1 March 2023).

Note that other stars and their planets also have magnetic activity. There are many astronomical observations (KEPLER, TESS and Evryscope-South Telescope), thanks to which large archives of observational data have been created. Stars have other modes of magnetic activity—for example, regular periods that are different from the solar, or without oscillations [36,37].

Thus, observations of sunspots and then magnetic fields on the Sun, which have been carried out since the beginning of the 20th century, have shown that the intensity of magnetic fields varies, and these changes are cyclical. At the beginning of the 11-year solar cycle, the large-scale solar magnetic field is directed predominantly along the meridians (it is commonly said that it is "poloidal") and has an approximate dipole configuration. At the maximum of the cycle, it is replaced by a magnetic field of sunspots directed approximately along the parallel (the so-called "toroidal"), which, at the end of the cycle, is again replaced by a poloidal one—while its direction is opposite to that observed 11 years ago ("Hale's law").

The solar dynamo model is intended to explain the mentioned observed features. Since the conductivity of the solar plasma is quite high, magnetohydrodynamics describes the magnetic fields in the Sun's convective zone. Due to the fact that the equatorial regions of the Sun rotate faster than the polar ones (this feature is called "rotation differential"), the initially poloidal field, being carried away by the rotating plasma, should stretch along the parallels, thereby acquiring a toroidal component. However, to ensure a closed, self-sustaining process, the toroidal field must somehow be transformed into a poloidal one. For some time, it was not clear how this happened. Moreover, Cowling's theorem explicitly forbade a stationary axisymmetric dynamo. In 1955, the American astrophysicist Eugene Parker, in his classic work [38], showed that the rising volumes of solar plasma

must rotate due to the Coriolis forces, and the toroidal magnetic fields entrained by them can be transformed into poloidal ones (the so-called "alpha effect"). Thus, a model of a self-sustaining solar dynamo was constructed.

Currently, numerous solar dynamo models that are more complex than Parker's have been proposed, but, for the most part, we revert back to the latter. In particular, it is assumed that the generation of magnetic fields does not occur in the entire convective zone of the Sun, as previously thought, but in the so-called "tachocline"—a relatively narrow region near the boundary of the convective and radiant zones of the Sun, at a depth of approximately 200,000 km under the solar photosphere, where the rotation speed changes sharply. The magnetic field created in this region rises to the surface of the Sun due to magnetic buoyancy.

The main dynamo models and their development are presented in the works [39,40].

Dynamo problems deal with the physical description of the process of generating a magnetic field by a conducting fluid. Field generation is based on turbulence, and turbulent dynamo models are divided into "large scale/medium dynamo" and "small scale/fluctuating dynamo." In the first group, magnetic fields are amplified on scales larger than the outer scale of turbulence in seconds on smaller scales [41]. In the second group, the small-scale turbulent dynamo is responsible for amplifying magnetic fields on scales smaller than the driving scale of turbulence in diverse astrophysical media [42].

In a turbulent dynamo, the amplification of magnetic fields occurs due to the turbulent stretching of magnetic fields (due to turbulent shear). The basic principle of generating a magnetic field was proposed by Zeldovich in the 20th century and called the Zeldovich "stretch-twist-fold" (STF) dynamo [43,44]. This is based on a simple principle, which is represented in Figure 3. The magnetic field in Figure 3 is represented as a closed rope, which, at the initial stage, is stretched while maintaining its volume, as in an incompressible flow (A → B in Figure 3)—for example, twice. In this case, the cross-section of the rope is reduced by a factor of two, and due to the freezing of the flow, the magnetic field increases by a factor of two. At the next stage, the rope is twisted into a figure "8" (B → C in Figure 3), and then folded (C → D in Figure 3) so that two loops appear, the fields of which are now directed in the same direction and together occupy the same volume as the original rope configuration. The flow through this volume has doubled. Then, the two loops merge into one (D → A in Figure 3) through small diffusion effects. Due to this, the process becomes irreversible. Such combined loops are topologically the same as the original single loop, but now with a factor of two field strength.

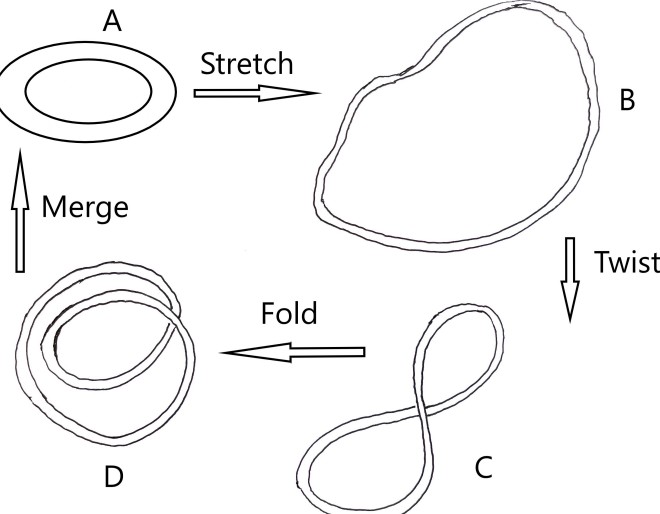

**Figure 3.** Sketch of Zeldovich "stretch-twist-fold" dynamo. A—initial state of magnetic field loop, B—stretched magnetic field loop, C—twisted loop, D—two loops are merged into one.

The earliest work on the turbulent dynamo theory was presented in [45,46] for the kinematic regime of the turbulent dynamo with a negligible back-reaction of magnetic fields. In [45], a simple model of the turbulent motion of a conducting liquid fluid was considered. The flow velocity has a Gaussian distribution function and the time for the establishment of diffusion of the liquid particles is zero. In this case, an exact solution of the problem of amplification of a spontaneous magnetic field can be derived. The instability criterion and magnetic field increment are obtained. In [46], the evolution of a weak, random initial magnetic field in a highly conducting, isotropically turbulent fluid is discussed with the aid of the exact expression for the initial growth of the magnetic energy spectrum. The possibilities of eventual growth and eventual decay are both admitted. For each, the shape of the magnetic-energy spectrum in the case $\lambda >> \nu$ ($\lambda$—magnetic diffusivity, $\nu$—kinematic viscosity) is estimated by simple dynamical arguments. If there is growth, it is concluded that the magnetic spectrum below the Ohmic cut-off eventually reaches equipartition with the kinetic energy spectrum, with the principal exception that the spectrum of kinetic energy in the equipartition inertial range evolves to the form $k^{-3/2}$ and that equipartition is maintained, with the rapidly falling spectrum, through part of the Ohmic dissipation range. The evolution of the magnetic spectrum in the weak-field $\lambda >> \nu$ regime is also computed numerically with a simplified transfer approximation suggested by the Lagrangian history direct interaction equations. This calculation is found to yield an eventual, very weak exponential growth in magnetic energy.

In the case wherein the magnetic back-reaction becomes significant, we have to consider the nonlinear turbulent dynamo. A nonlinear turbulent dynamo is characterized by energy equipartition between turbulence and magnetic fields within the inertial range of turbulence [47].

In [48], kinematic dynamo theory is presented for turbulent conductive fluids. The authors described how inhomogeneous magnetic fluctuations are generated below the viscous scale of turbulence, where the spatial smoothness of the velocity permits a systematic analysis of the Lagrangian path dynamics. In [48], the authors found the magnetic field's moments and multipoint correlation functions analytically at small yet finite magnetic diffusivity. The authors showed that the field is concentrated in long, narrow strips and described anomalous scalings and angular singularities of the multipoint correlation functions, which are manifestations of the field's intermittency. The growth rate of the magnetic field in a typical realization is found to be half the difference of two Lyapunov exponents of the same sign.

In [49], the authors showed the results of an extensive numerical study of the small-scale turbulent dynamo. The primary focus is on the case of large magnetic Prandtl numbers $P_m$, which is relevant for hot, low-density astrophysical plasmas. A $P_m$ parameter scan is given for the model case of viscosity-dominated (low Reynolds number *Re*) turbulence. The authors concentrated on three topics: magnetic energy spectra and saturation levels, the structure of the magnetic field lines and the intermittency of the field strength distribution. In [49], the main results are as follows: (1) the folded structure of the field (direction reversals at the resistive scale, field lines curved at the scale of the flow) persists from the kinematic to the nonlinear regime; (2) the field distribution is self-similar and appears to be lognormal during the kinematic regime and exponential in the saturated state; and (3) the bulk of the magnetic energy is at the resistive scale in the kinematic regime and remains there after saturation, although the magnetic energy spectrum becomes much shallower. The authors proposed an analytical model of saturation based on the idea of partial two-dimensionalization of the velocity gradients with respect to the local direction of the magnetic folds. The model-predicted saturated spectra are in excellent agreement with the numerical results. Comparisons with large-*Re*, moderate-$P_m$ runs are carried out to confirm these results' relevance and test heuristic scenarios of dynamo saturation. New features at large *Re* are the elongation of the folds in the nonlinear regime from the viscous scale to the box scale, and the presence of an intermediate nonlinear stage of slower than exponential magnetic energy growth accompanied by an increase in the resistive scale

and partial suppression of the kinetic energy spectrum in the inertial range. Numerical results for the saturated state do not support scale-by-scale equipartition between magnetic and kinetic energies, with a definite excess of magnetic energy at small scales. A physical picture of the saturated state is proposed.

In [47,50], the authors described the striking similarity between the dependence of dynamo behavior on Prandtl number $P_m$ in a conducting fluid and $R$ (a function of ionization fraction) in the partially ionized gas. In a weakly ionized medium, the kinematic stage is largely extended, including exponential growth and a new regime of dynamo characterized by the linear-in-time growth of the magnetic field strength, and the resulting magnetic energy is much higher than the kinetic energy carried by viscous-scale eddies. Unlike the kinematic stage, the subsequent nonlinear stage is unaffected by microscopic diffusion processes. It has a universal linear-in-time growth in magnetic energy with the growth rate as a constant fraction 3/38 of the turbulent energy transfer rate, which agrees well with earlier numerical results. Applying the analysis to the first stars and galaxies, S. Xu [50] found that the kinematic stage can generate a field strength only an order of magnitude smaller than the final saturation value. However, the generation of large-scale magnetic fields can only be accounted for by the relatively inefficient nonlinear stage and requires a longer than free-fall time. This suggests that magnetic fields may not have played a dynamically important role during the formation of the first stars.

Figure 4 illustrates the magnetic energy spectrum in the nonlinear stage of the turbulent dynamo. At $P_m = 1$, it follows the Kazantsev $k^{3/2}$ profile on scales larger than $1/k_p$, while, on smaller scales, the transition to MHD turbulence occurs, and there is a $k^{-5/3}$ range for both the kinetic and magnetic energies.

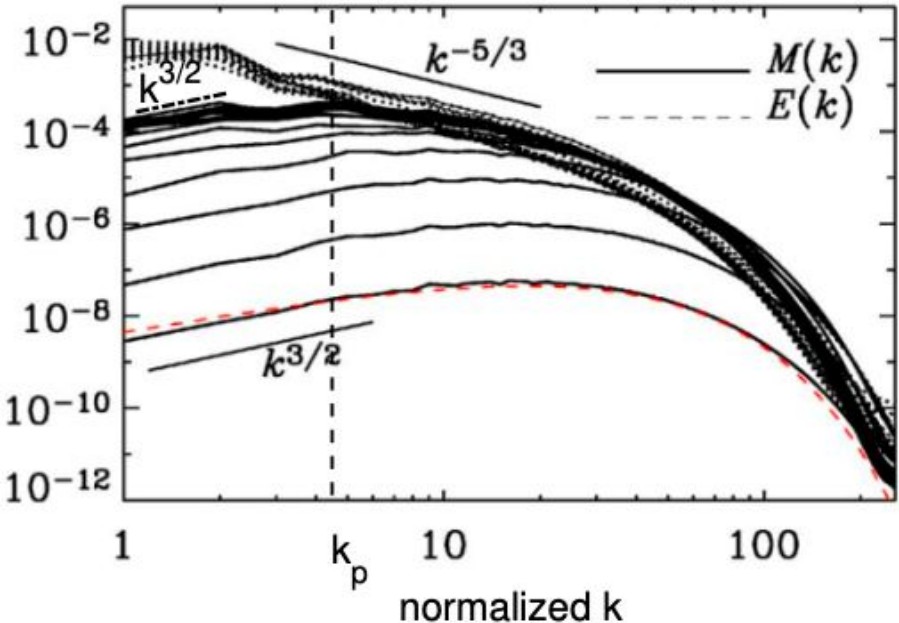

**Figure 4.** Sketch of the magnetic (solid line) and turbulent kinetic (dashed line) energy spectra in the nonlinear stage of a turbulent dynamo for $P_m = 1$; dash-dotted lines indicate different spectral slopes and the vertical dashed line represents the equipartition scale at the end of their simulations. Turbulent kinetic (red dashed line) energy spectra in the nonlinear stage of a turbulent dynamo for $P_m = 1$. From [50].

Thus, the general idea of the emergence of the turbulent dynamo process can be outlined in a diagram, as shown in Figure 5.

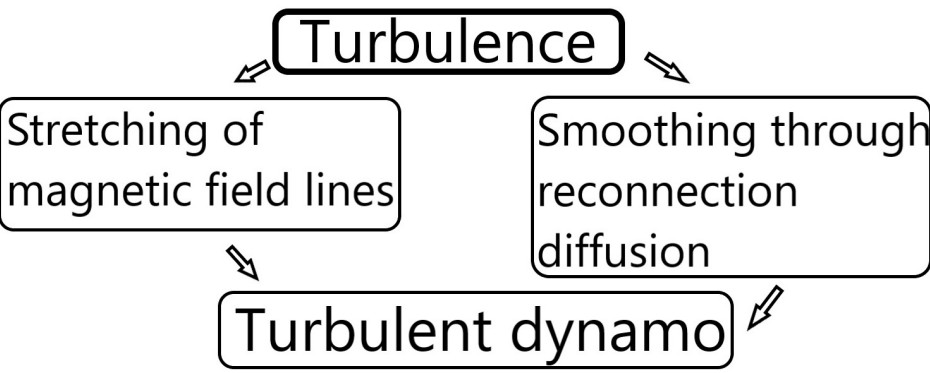

**Figure 5.** Scheme of the turbulent dynamo process.

## 4. Magnetic Reconnection and Turbulence

### 4.1. Fast Turbulent Reconnection

Turbulence accelerates many transport processes, e.g., those of heat and mass diffusion. The quantitative theory of turbulent reconnection was formulated in [21]. The current state of this theory and its implications are summarized in [51].

In laminar fluids with high conductivity, the magnetic fields are frozen in the fluid. This means that the motion of the fluid induces the motion of the magnetic field line. The annihilation of magnetic field lines happens when the field lines of opposite directions come into close contact. This situation is shown in the upper panel of Figure 6. The reversing component of upper and lower magnetic fluxes is shown, while the shared magnetic field is perpendicular to the plane of the cartoon. This component is passive, and it does not participate in reconnection.

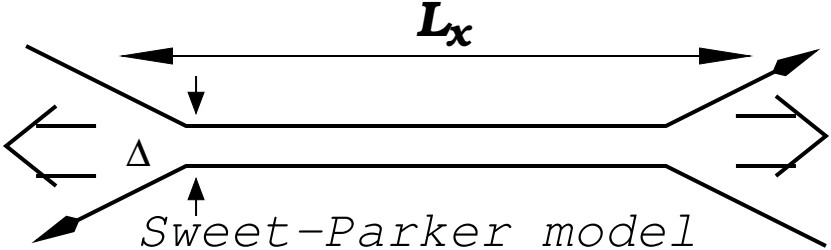

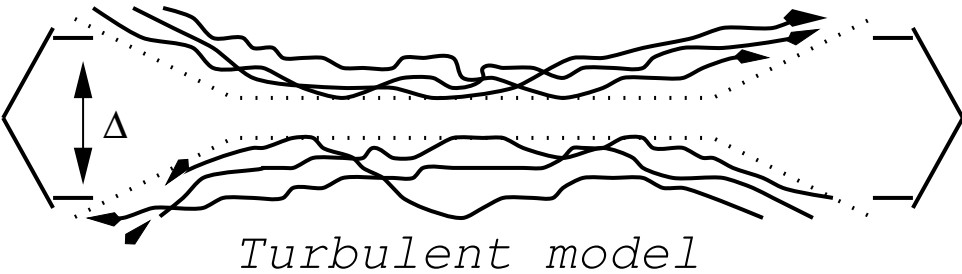

**Figure 6.** *Upper plot*: Classical Sweet–Parker (SP) model of reconnection: the thickness of the outflow $\Delta$ is limited by Ohmic diffusivity. $L_x \gg \Delta$ makes the SP reconnection slow. *Lower plot*: Ref. [21]'s model of reconnection includes turbulence in the SP setting. The outflow width $\Delta$ is determined by macroscopic field line wandering, and it can be $\sim L_x$ for trans-Alfvénic turbulence. From [21].

The low rates of magnetic reconnection follow from the geometry of the Sweet–Parker configuration. The reconnection can be fast only if the magnetic field lines come close

together, so Ohmic resistivity can annihilate them. This means that the distance between the lines $\Delta$ should be small. However, the fluid that carries the magnetic field lines leaves the system for this to happen. The outflow happens with Alfvén velocity through the slot $\Delta$. If $\Delta$ is small, the reconnection speed $V_{rec} \approx V_A \Delta / L$ is small. If $\Delta$ is large, the reconnection stops as the magnetic field lines do not come close to each other to reconnect. As a result, the Sweet–Parker reconnection is negligibly small in astrophysical conditions.

Turbulence introduces magnetic field meandering. The thickness of the outflow $\Delta$ is determined by this meandering. At the same time, the magnetic field lines can occasionally come close together, enabling fast reconnection to happen locally. In other words, turbulence resolves the contraditiction of the Sweet–Parker model, i.e., the magnetic fields are close together for fast Ohmic local reconnection and the magnetic fields must be far apart for the thick outflow to exist.

The model in [21] of turbulent reconnection is presented in Figure 6. It is a natural generalization of the classical Sweet–Parker model of reconnection in the case of turbulence. Both fluxes share a magnetic field of the same direction perpendicular to the figure's plane. The value of this so-called "guide field" does not change the reconnection rate (see [51]). This magnetic field component is ejected from the reconnection region with plasmas/fluid.

Unlike other models of fast magnetic reconnection, the [21] model does not appeal to plasma effects but accounts for the magnetic field wandering induced by Alfvénic turbulence. Employing the Alfvénic mode scaling that we presented in §4, [21] obtained the expression for the reconnection rate in the turbulence with injection at the scale $L$ and the opposite magnetic field in contact over scale $L_x$:

$$V_{rec} \approx V_A \min\left[ \left(\frac{L_x}{L}\right)^{1/2}, \left(\frac{L}{L_x}\right)^{1/2} \right] M_A^2. \tag{9}$$

Note that $V_A M_A^2$ is proportional to the turbulent eddy speed. As we discussed earlier, the obtained reconnection rate varies depending on the Alfvén Mach number $M_A$ and for $M_A \sim 1$ can represent a large fraction of the Alfvén speed.

It is clear from Equation (9) that the turbulent reconnection rate can be both slow and fast, depending on the system's turbulence level. This allows the reconnection model to explain various energetic phenomena that are impossible for the given prescribed reconnection rate. For instance, to explain the explosive release of magnetic energy in solar flares and gamma-ray bursts (see [52]), it may be necessary to have periods of slow reconnection to accumulate the flux and periods of fast reconnection during which the energy is being released. The [21] model was tested numerically ([53,54]) and compared with observations ([55,56]). In [57], the authors carried out a statistical study of flaring active regions that produced strong solar flares of an X-ray class X1.0 and higher during a time period that covered solar cycles 23 and 24. It was found that in 72 percent of cases, the flaring active regions did not comply with the empirical laws of the global dynamo, and it appears that the flaring is governed by the turbulent component of the solar dynamo. These observational findings are in consensus with the concept of the essential role of nonlinearities and turbulent intermittence in the magnetic field generation inside the convective zone, which follows from dynamo simulations.

It is important that the outflow from the reconnection zone can induce turbulence, making turbulent reconnection self-induced. This process was successfully tested in a number of numerical studies [58–61]. Figure 7 shows that the spectrum of turbulence induced by magnetic reconnection evolves towards the Kolmogorov-type cascade of Alfvénic turbulence.

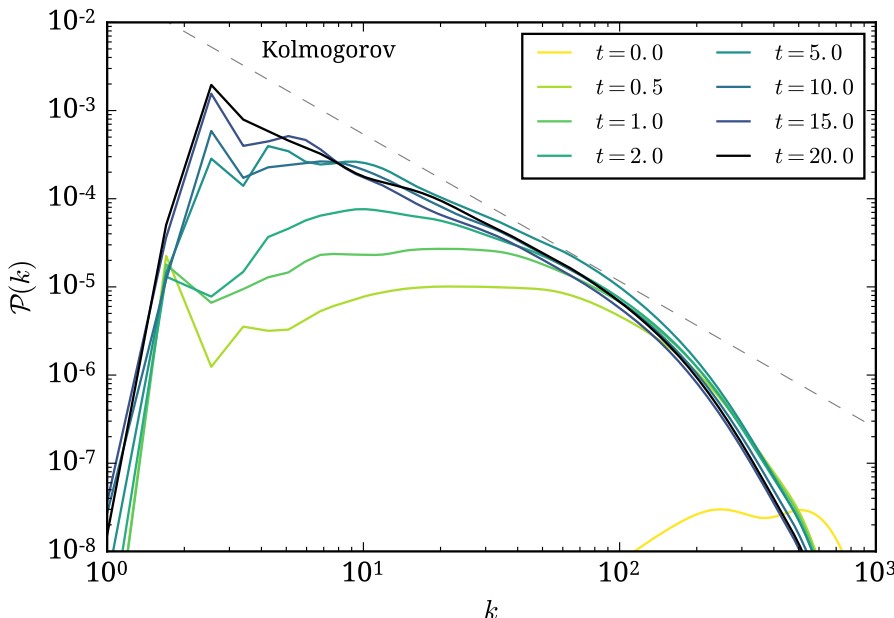

**Figure 7.** Evolution of turbulent velocity power spectra of the self-induced reconnection. From [61].

Turbulent reconnection induces particle acceleration ([62]). The predicted acceleration was successfully tested in [63].

### 4.2. Violation of Flux Freezing and Reconnection Diffusion

The theorem of Alfvén (1942) predicts that magnetic flux is frozen within a conductive fluid, i.e., the magnetic field and plasma move together. Turbulent reconnection violates the flux freezing condition. It was suggested in [62] that in a turbulent fluid, the magnetic field decouples from the gas and can diffuse, solving, for instance, the problem of magnetic flux in star formation. More theoretical studies in [22] support this idea. The flux freezing violation was demonstrated numerically in [64].

Fast turbulent reconnection allows the efficient exchange of magnetic fields and plasmas between adjacent turbulent eddies. This is illustrated in Figure 8, where the interaction of eddies of a given scale is shown. In reality, such interactions proceed at every scale of turbulent motion, which ensures the efficient diffusion of magnetic fields.

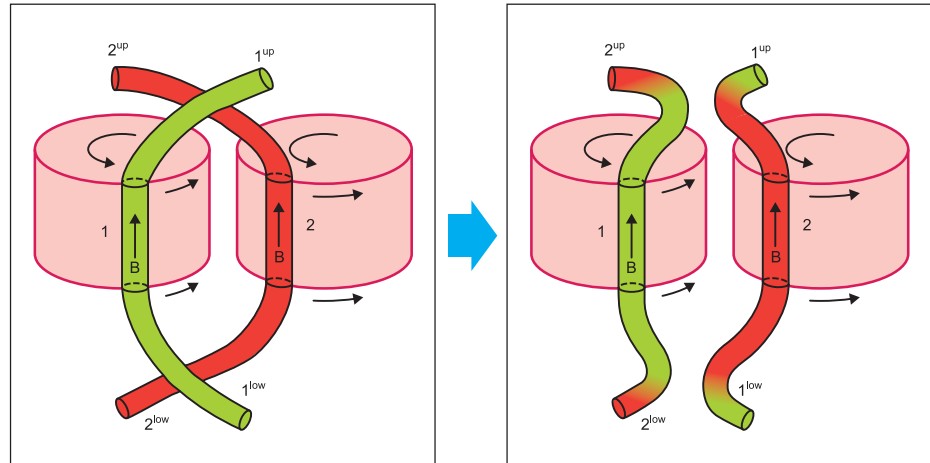

**Figure 8.** Illustration of reconnection diffusion. Matter and magnetic fields are exchanged as two flux tubes of adjacent eddies interact. From [65], © AAS. Reproduced with permission.

## 5. Turbulence in Spiral Galaxies

### 5.1. Properties of Interstellar Turbulence

The interstellar media of spiral galaxies are turbulent. Figure 9 presents the spectra of electron density fluctuation and velocity fluctuation that are observed in the Milky Way. Both power laws correspond to the Kolmogorov scaling, which has a natural explanation within MHD turbulence theory.

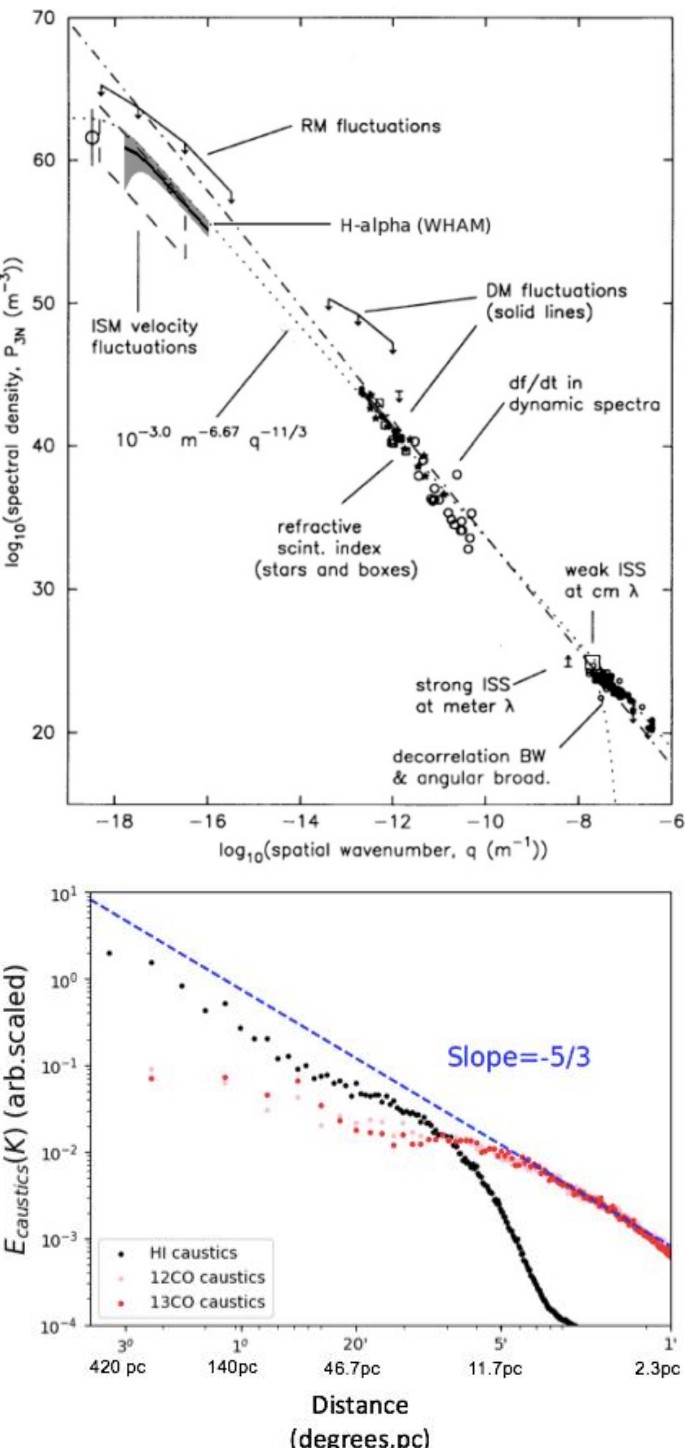

**Figure 9.** *Upper panel:* "Extended Big Power Law" of galactic electron density fluctuations obtained combining the scattering measurements in [1] and $H_\alpha$ measurements from WHAM in [2]. *Lower panel:* Power law of *velocity* fluctuations in the direction of the Taurus molecular cloud. From [66].

The fact that the velocity spectrum is Kolmogorov reflects the Kolmogorov scaling of perpendicular motions in Alfvénic turbulence. These motions dominate the observed cascade. As for the density fluctuations, the weakly compressible turbulence corresponding to the warm galactic gas passively reflects the statistics of velocity fluctuations. Note that for compressible parts of the media, steeper-density spectra were observed see [67].

*5.2. Effects on Star Formation*

Turbulence effects are essential for star formation. MHD turbulence has been used for classical theories of star formation as a supporting molecular cloud from the gravitational collapse. Such an approach requires long-lived turbulent motions. Thus, the authors appealed to magnetic fields to mitigate turbulence decay. Alfvénic turbulence, as discussed above, dissipates in one eddy turnover time. Thus, unless strong internal sources of turbulence driving exist, turbulence is hardly a means of preventing the typical molecular clouds from collapsing along magnetic field lines.

However, turbulence is successful in an unexpected task. In the star formation theory, a major issue is the removal of the magnetic fields from collapsing clouds. In magnetically mediated star formation theory (see [68–74]), magnetic fields counteract gravitational collapse. The magnetic flux freezing is assumed to be frozen within the ionized component, while the change in the flux-to-mass ratio is due to neutrals that flow past ions and are concentrated towards the center of the gravitational potential. In a sense, ions act as guards that obey the magnetic field, while neutrals percolate through their ranks, experiencing viscosity due to neutral–ion collisions. The latter process is termed in the star formation literature "ambipolar diffusion". For decades, ambipolar diffusion was assumed to be the necessary condition for star formation in the ISM.

During ambipolar diffusion, the magnetic field resists the compression and leaves the gravitational potential, while neutrals are concentrated, forming the propostar (e.g., [75,76]). The mediation of ambipolar diffusion was assumed to make star formation inefficient for magnetically dominated (i.e., subcritical) clouds. The slow speed of ambipolar diffusion entails low efficiency of star formation, which was interpreted as strong observational support of the ambipolar diffusion paradigm (e.g., [77]).

This, however, does not solve all the problems as, for clouds dominated by gravity, i.e., supercritical clouds, the magnetic fields do not have time to leave the cloud through ambipolar diffusion. Therefore, for such clouds, magnetic fields are expected to be dragged into the star, forming stars with magnetizations far in excess of the observed ones (see [78,79]).

The process of reconnection diffusion illustrated by Figure 8 provides a viable explanation of the magnetic flux removal processes in the star formation process. The analytical prediction of the reconnection diffusion rate was confirmed numerically in [80]. This process induces fast magnetic flux removal independent of the media ionization degree.

The application of reconnection diffusion allows us to solve the long-standing problem of magnetic flux removal from accretion disks, the so-called "magnetic braking catastrophe". The essence of the problem is that during circumstellar disk formation, the magnetic fields of molecular clouds are able to transfer the matter momentum from the forming disk on a time scale shorter than the disk formation time. Figure 10, from numerical studies in [81], shows that the problem can be solved if reconnection diffusion is accounted for. The authors plot the results of the simulations of disk formation in a magnetized interstellar medium. Without turbulence, there is no reconnection diffusion. Thus, magnetic flux freezing transforms ensure that the angular momentum of the collapsing material is transported out of the disk. The resulting disks are too small and do not correspond to observations. In the presence of turbulence, the magnetic field diffuses from the disk due to reconnection diffusion. The magnetic coupling of the material of the disk and the ambient interstellar medium is reduced. As a result, there is no catastrophic loss of the angular momentum of the disk. The resulting disks are larger and correspond to the observed ones.

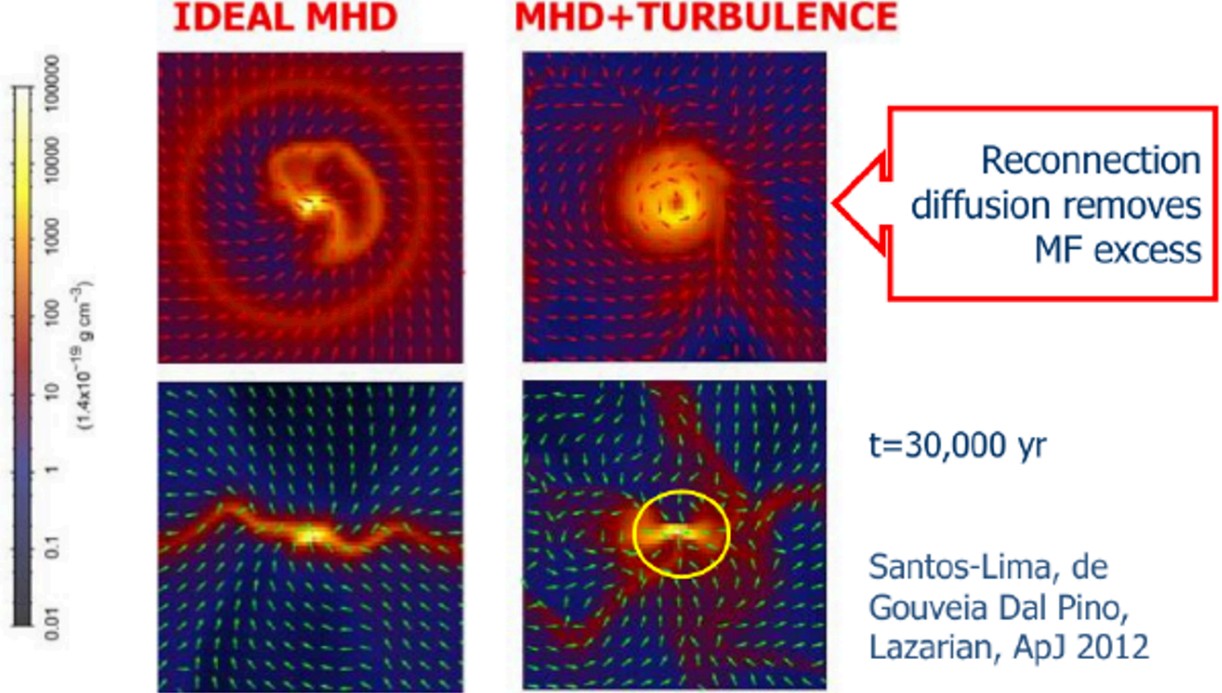

**Figure 10. Left**: Evolution of disk without reconnection diffusion. The formed disk is much smaller compared to observed ones. **Right**: The disk produced via reconnection diffusion in 30,000 years. From [81].

## 6. Turbulence and Cosmic Rays

Particles with energies ranging from MeV to PeV are usually termed cosmic rays (CRs). Their interaction with MHD turbulence controls the CR diffusion; their acceleration also depends on this interaction [82]. The knowledge of CR propagation is vital for understanding the solar modulation of galactic CRs, Fermi bubble emission and space weather forecasting [83–85]. It is also essential for an understanding of the origin of driving for galactic winds (e.g., [86,87]) and feedback heating in clusters of galaxies (e.g., [88,89]).

CRs can interact with the pre-existing MHD fluctuations and the magnetic fluctuations created by them, e.g., by the perturbations created by the streaming instability (see [90]). The suppression of streaming instability by MHD turbulence [91–94] can significantly modify the CR propagation [95]. In [96], the authors considered the propagation of cosmic rays in turbulent magnetic fields using the models of magnetohydrodynamic turbulence that were tested in numerical simulations, in which the turbulence is injected on large scales and cascades to small scales.

Earlier, the CRs' interaction with magnetic turbulence was studied with ad hoc models adopted for MHD turbulence [90,97–99]. These involved the model of isotropic MHD turbulence (see [100]), as well as the 2D + slab model for solar wind turbulence [101]. These, however, do not correspond to the understanding of the modern MHD turbulence presented in Section 2.

For CR propagation, distinguishing the propagation perpendicular to the mean magnetic field and the propagation parallel to the mean magnetic field is useful. The perpendicular propagation is governed by magnetic field wandering as described in [21]. This induces processes of diffusion and superdiffusion, as illustrated in Table 1 from [102]. Table 1 shows the main scales in turbulent processes. Power laws are different for subAlfvénic turbulence in the weak and strong regimes, but it is the same Kolmogorov one for superAlfvénic turbulence. Thus, for superAlfvénic turbulence, there is only one power law with $k_\perp^{-5/3}$; for subAlfvénic turbulence, there is a broken power law with $k_\perp^{-2}$ weak turbulence and $k_\perp^{-5/3}$ at strong MHD turbulence.

**Table 1.** Regimes of MHD turbulence and magnetic diffusion.

| Type of MHD Turbulence | Injection Velocity | Range of Scales | Spectrum E(k) | Motion Type | Ways of Study | Magnetic Diffusion | Squared Separation of Lines |
|---|---|---|---|---|---|---|---|
| Weak | $V_L < V_A$ | $[l_{trans}, L]$ | $k_\perp^{-2}$ | wave-like | analytical | diffusion | $\sim sLM_A^4$ |
| Strong subAlfvenic | $V_L < V_A$ | $[l_{min}, l_{trans}]$ | $k_\perp^{-5/3}$ | anisotropic eddy-like | numerical | Richardson | $\sim \frac{s^3}{L}M_A^4$ |
| Strong superAlfvenic | $V_L > V_A$ | $[l_A, L]$ | $k_\perp^{-5/3}$ | isotropic eddy-like | numerical | diffusion | $\sim sl_A$ |
| Strong superAlfvenic | $V_L > V_A$ | $[l_{min}], l_A$ | $k_\perp^{-5/3}$ | anisotropic eddy-like | numerical | Richardson | $\sim \frac{s^3}{L}M_A^3$ |

$L$ and $l_{min}$ are the injection and perpendicular dissipation scales, respectively. $M_A \equiv \delta B/B$, $l_{trans} = LM_A^2$ for $M_A < 1$ and $l_A = LM_A^{-3}$. for $M_A < 1$. For weak Alfvenic turbulence $\ell_\parallel$ does not change. $s$ is measured along magnetic field lines.

The CR superdiffusion relates the transposition of the CR along the magnetic field to the transposition in the perpendicular direction. The accelerated diffusion of CRs in the perpendicular direction is illustrated by Figure 11. The superdiffusion acts on scales less than the turbulence injection scale, radically changing the CR dynamics.

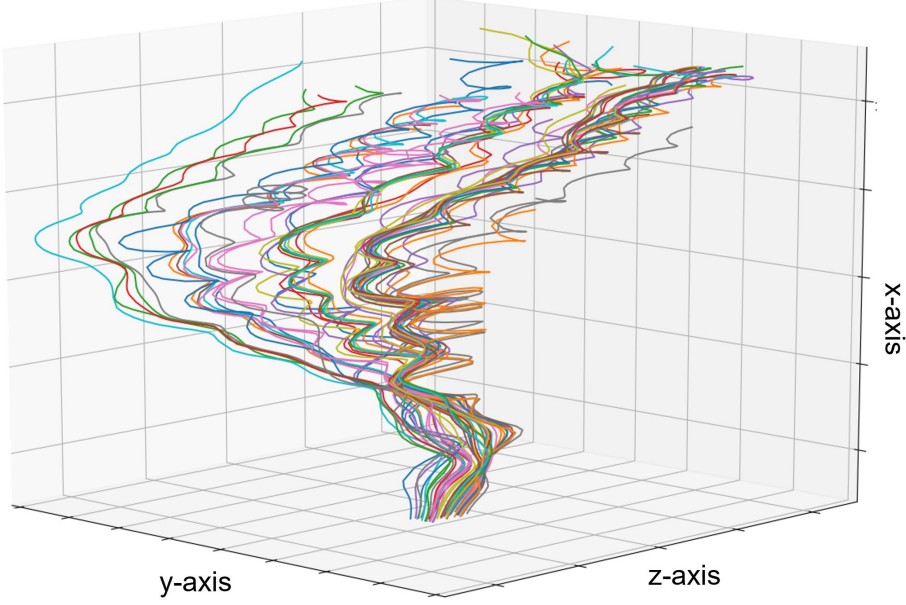

**Figure 11.** CRs' superdiffusion: CRs' trajectories are shown with $M_S = 0.62$ and $M_A = 0.56$. The initial spatial separation between CRs is one pixel and the initial pitch angle is 0 degrees. From [20].

In terms of parallel to magnetic field diffusion, pitch angle scattering and Transient Time Damping (TTD) are generally accepted processes [100]. The first process arises from the resonant scattering of particles with both compressible and incompressible fluctuations. In [103], pitch angle scattering by fast modes was identified as the dominant process of scattering. The second is from particles surfing the compressible magnetic fluctuations, which, in many cases, arise from particles surfing slow modes [104]. Understanding the CR perpendicular superdiffusion allowed [105] to introduce a new process termed *bouncing diffusion*. This type of diffusion arises from the simultaneous action of reflection from magnetic mirrors induced by slow and fast modes and the perpendicular superdiffusion. The bouncing diffusion acts on particles with small pitch angles; this slows their diffusion. A comparison of the diffusion coefficients for bouncing and non-bouncing particles is provided in Figure 12.

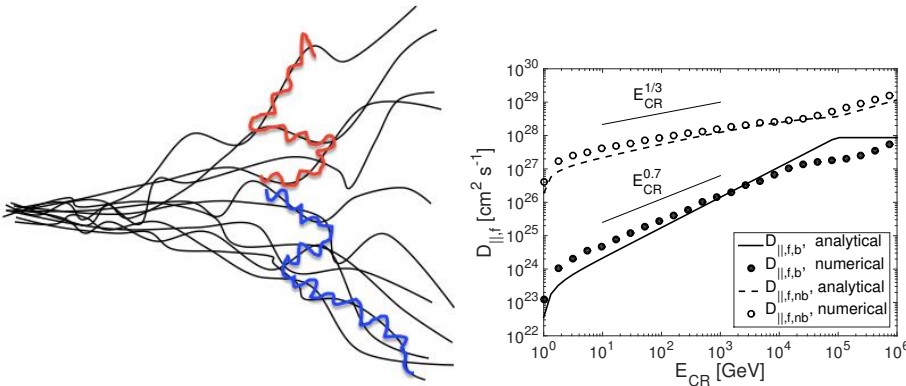

**Figure 12. Left**: Schematic of bouncing diffusion. Trajectories of two particles with small initial separation are shown. **Right**: The comparison of the parallel diffusion coefficients induced by fast modes $D_{\|,f,b}$ of bouncing CRs and $D_{\|,f,nb}$ of non-bouncing CRs. From [105].

With fast modes providing efficient isotropization of CRs in terms of pitch angle $\mu$, ref. [105] evaluated the total distribution of scattering and bouncing particles as

$$D_{\|,\text{tot}} \approx \mu_c^2 D_{\|,b} + (1 - \mu_c^2)D_{\|,nb}, \tag{10}$$

where $D_{\|,scat}$ is the diffusion coefficient arising from scattering. The bouncing diffusion prevents the fast escape of particles with $\mu < \mu_{cr}$. Individual particles can generally exhibit periods of slow bouncing diffusion separated by periods of fast diffusion when they are in the scattering regime, i.e., Levi flights.

## 7. Implication of Turbulence: Gradient Technique for Studies of Magnetic Fields

Studies of astrophysical magnetic fields rely on the effects of the magnetic field on the media. Mapping of plane-of-sky magnetic fields can be obtained with polarization from grains aligned with long axes perpendicular to the magnetic field [106], and synchrotron polarization [107].

The above techniques employ polarization, and polarization measurements require significantly more effort than measurements of signal intensities. Therefore, the Gradient Technique (GT), which allows the mapping of the plane-of-sky (POS) magnetic field without polarization measurements, opens up a new avenue for magnetic field studies in diffuse media. The GT employs the properties of magnetic turbulence. The versions of the technique that do not require polarization measurements have been implemented as the *Velocity Gradient Technique (VGT)* with subdivision of Velocity Centroid Gradients (VCGs) that employ Velocity Centroids ([108,109]) and Velocity Channel Gradients (VChGs) [110] that employ intensity fluctuations in thin channel maps, as well as *Synchrotron Intensity Gradients (SIGs* [111] that employ synchrotron intensities. Note that the GT can also employ polarization to obtain extra information about the magnetic field. For instance, as shown in [112], Synchrotron Polarization Gradients (SPGs) can use synchrotron polarization at different wavelengths to probe magnetic fields at different distances along the line of sight (see [113]), while Faraday Gradients (FGs) can obtain the distribution of the plane-of-sky direction of the magnetic field. However, we do not discuss polarization versions in the present paper.

Due to MHD turbulence's properties, as discussed in Section 3, the eddies are aligned with the magnetic field. The eddies' rotation along the magnetic field's local direction induces velocity and magnetic field gradients perpendicular to the magnetic field. In Kolmogorov-type turbulence of Alfvénic and slow modes, the gradients increase with the decrease in eddy scale as $v_l/l_\perp \sim l_\perp^{-2/3}$. Thus, the smallest resolved eddies well aligned with the magnetic field dominate the gradients. Consequently, the gradients are perpendicular to the local direction of the magnetic field, revealing the magnetic field structure.

Figure 13 demonstrates the power of Synchrotron Gradients (SIGs). The maps of the magnetic field obtained with Planck synchrotron polarization are compared to those obtained with SIGs. In most cases, the difference in the obtained directions is negligible. A major advantage of SIGs is that they are not subject to Faraday rotation distortions. These are especially harmful for low frequencies.

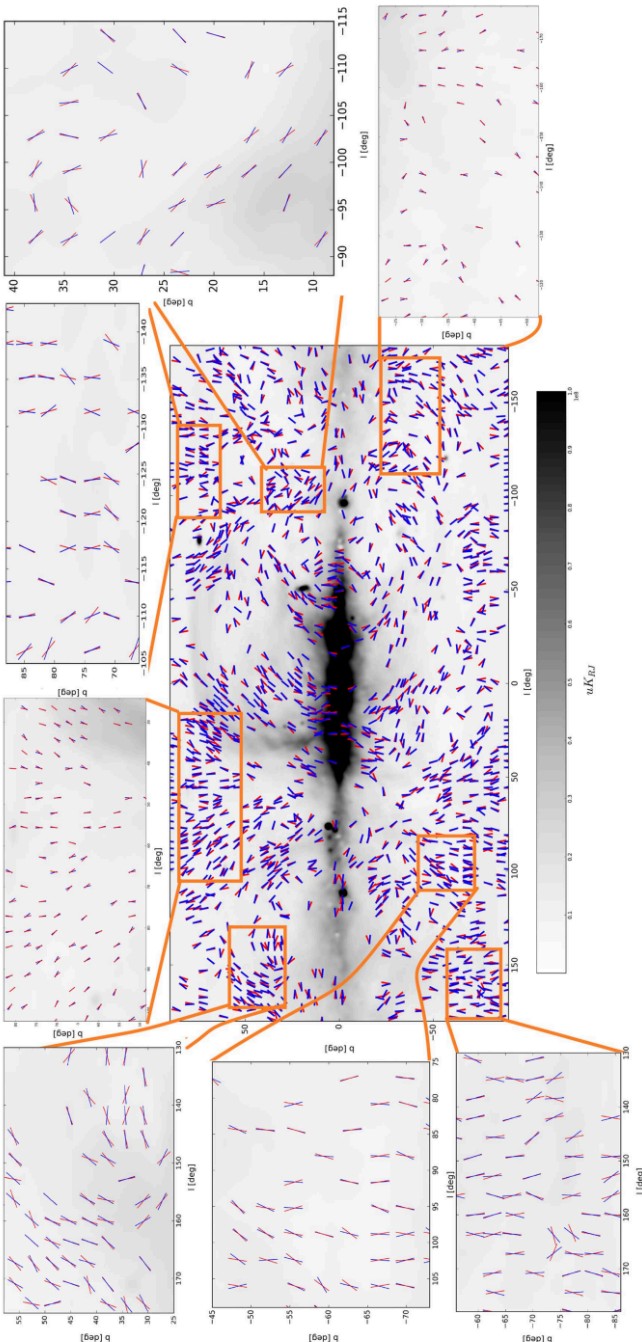

**Figure 13.** Comparison of galactic magnetic fields obtained with polarization and SIGs. From [111].

Figure 14 shows the magnetic field structure of the active galaxy M51 mapped by Velocity Gradients (VGT) and synchrotron polarization. The structure of the galactic magnetic field is better resolved with gradients compared to VLA synchrotron polarization measurements. Comparing the magnetic field maps obtained with the VGT and dust polarization in [114] reveals important details of the effects of the magnetic field on the central black hole accretion.

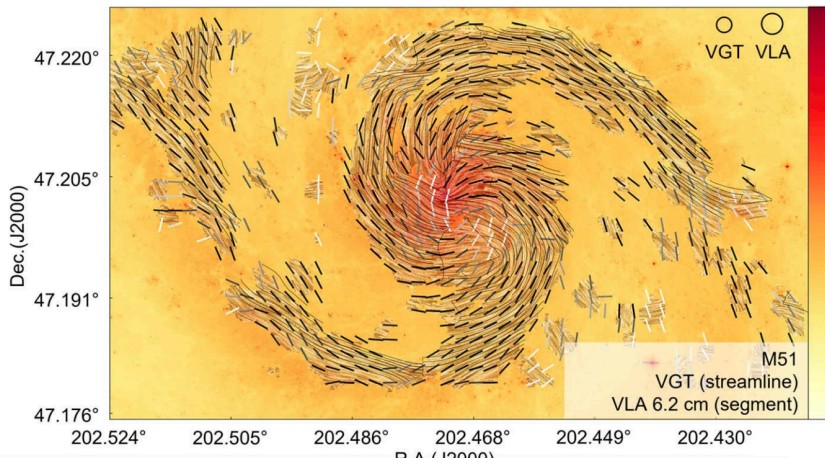

**Figure 14.** Velocity gradients obtained from publicly available $^1 2CO$ maps from ALMA provide better-resolution magnetic field maps compared to VLA. From [114].

The GT can provide magnetic fields in situations where all other techniques fail. For instance, in [115], the POS magnetic fields were mapped with VChGs for a tenuous high-velocity Smith cloud, which would be impossible with any other existing technique. Figure 15 presents a similar case where the unique abilities of gradients allow magnetic field studies of galaxy clusters. The Chandra X-ray emission is used for mapping. For subsonic turbulence, the density inhomogeneities that control X-ray emission mimic velocity fluctuations. Thus, the Intensity Gradients (IGs) act similarly to velocity gradients.

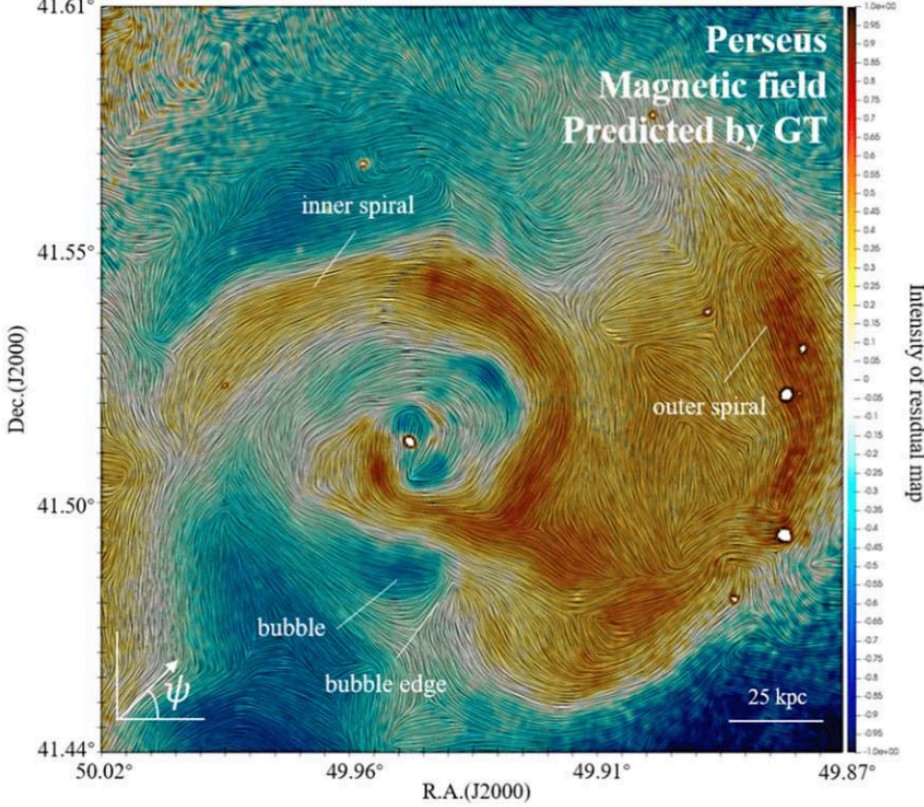

**Figure 15.** Velocity gradients obtained from publicly available $^1 2CO$ maps from ALMA provide a better-resolution magnetic field map compared to VLA. From [114].

## 8. Discussion

This review provides a brief outlook on the modern state of the theory of MHD turbulence and its implications. The theory of MHD turbulence is an area of intensive research, and new discoveries of subtle turbulence properties are expected. However, despite the available advances, the MHD turbulence theory is a powerful tool for exploring astrophysical processes.

We have discussed a wide variety of processes radically changed by turbulence. These include cosmic ray propagation and acceleration, star formation and dynamo. This review does not focus in depth on the particular applications. Instead, it provides a guide for researchers interested in the various astrophysical applications of MHD turbulence theory. A more thorough study is possible with the original papers and focused reviews that we refer to.

In the review, we emphasize the intrinsic, deep connection between the theory of MHD turbulence and the theory of turbulent reconnection. The fundamental properties of MHD turbulence, e.g., magnetic field wandering, cannot be understood without understanding turbulent magnetic reconnection. In fact, it is the turbulent reconnection that makes the description of MHD turbulence self-consistent. In magnetized turbulent fluids, turbulent reconnection is responsible for a process of *reconnection diffusion* that removes magnetic flux from molecular clouds and resolves the problem of the catastrophic breaking of matter in circumstellar disks.

To exemplify the power of MHD turbulence theory, we have discussed the technique of magnetic field study that utilizes the properties of MHD turbulence; specifically, the turbulent reconnection is a part of the MHD turbulent cascade. This new technique, called the Gradient Technique (GT), has proven to be a powerful tool for studying magnetic fields in the Milky Way, nearby galaxies and galaxy clusters.

**Author Contributions:** E.P. and A.L. made equal contributions. All authors have read and agreed to the published version of the manuscript.

**Funding:** A.L. acknowledges the support of NASA ATP AAH7546.

**Data Availability Statement:** Not applicable.

**Conflicts of Interest:** The authors declare no conflict of interest.

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
