# Peer review of "Outlook on Magnetohydrodynamical Turbulence and Its Astrophysical Implications"

_fluids, doi:10.3390/fluids8050142_

Round 1

Reviewer 1 Report

Review Report on the paper(magnetochemistry-2262875):

 “Outlook on Magnetohydrodynamical Turbulence and its Astrophysical Implications”

by Alexandre Lazarian, Elena Popova 

   This paper introduces the current status and significance of MHD turbulence theory. It focuses on a wide variety of processes that are radically altered by turbulence, including cosmic ray propagation and acceleration, star formation, and dynamo. And this paper emphasizes the deep connection between MHD theory and turbulence reconnection theory. In order to illustrate the power of MHD turbulence theory, the gradient technique (GT) for magnetic field study using MHD turbulence properties is discussed. This technique is of great significance in the study of the magnetic fields of the Milky Way, nearby galaxies and galaxy clusters.

   Magnetohydrodynamic (MHD) turbulence is ubiquitous in magnetized astrophysical plasmas and fundamentally alters a wide variety of astrophysical processes. This paper also gives the concept of MHD turbulence and explains the origin of its scaling, and considers the effects of MHD turbulence on a variety of problems: generators in different types of stars, flare activity, solar and stellar winds from different stars, cosmic ray propagation, and star formation.

   However, there are still many technical points and misprints. Hence, I suggest minor repairs.

Technical points and misprints:

1. Page 1, line 17: “through the numerous....... The first letter of the sentence should be capitalized. Please check it carefully.

2. The title in Figures 3,5,7,10,13,14 lacks punctuation at the end. Please check them carefully.

3. Page 14: The content in Figure 9 is not clear and the picture is separated from the title. Please revise it.

4. Page 14, line 11: Punctuation is wrong. Please check and correct it carefully.

5. Page 19, lines 28 and 30: Missing punctuation, please correct.

6. Page 5, line 15: ......new simulations shown in Fig. 5.0.1 suggest that the spectrum.......  Where is Fig.5.0.1 in the article?

7. Page 7: In[45]....... Please keep the format consistent with the rest of the article. Add space.

8. The use of "Fig." and "Figure" is recommended in the same way.

9. The section 5 of this article has only one section 5.0. It is recommended to change the section number for this section.

10. Page 12, line 13: ......(see [68],[69],[70],[71],[72],[73],[74])...... . We propose to write ......(see [68-74]).......

11. Article content and layout can be more organized. Especially the placement of pictures. Please also carefully check the article existing similar to the above problems.

Author Response

Thank you very much for your comments. We've made corrections. The final layout of the paper will be done by the publisher, we used the publisher's standard template.

Reviewer 2 Report

This manuscript reviews the physics of the MHD turbulence and summarizes its applications in multiple circumstances in Astrophysics. As a review paper, this manuscript does the job; however, without any initial research results reported, I do not consider this manuscript is a research article. Whether I recommend this manuscript for publication or not, depends on the journal's need. Minor concerns are: some inserted text or figures are too small to read, for example, Table 1, Figure 7 and Figure 12. 

Author Response

Thank you for your comment. This is the review paper that a publisher asked us to do. We made Table 1, Figure 7, and Figure 12 bigger.

Reviewer 3 Report

please see attachment.

Author Response

Thank you very much for this comment. We added the reference to the Chapter "Magnetic reconnection and turbulence".

Reviewer 4 Report

Review of article "Outlook on Magnetohydrodynamical Turbulence and its Astrophysical Implications" by E. Popova and A. Lazarian.

This is an interesting review article on MHD turbulence that would be interesting to the astronomical community as well as readers who are interested in in turbulence and MHD in general. (given the journal scope).

The article in my opinion has a lucid style, but requires moderate English language changes due typo/grammatical errors.

# The article description is primarily driven by scaling analysis which is the backbone of turbulence theory. The authors have done a good job at explaining theory without invoking equations and just using scaling laws. I think the readers would benefit if the authors can provide an infographic/schematic of MHD turbulence with all the relevant length scales involved in MHD. They do something in Table 1 (printed as Image 9).

# I had to read the Dynamo section twice to make sure I am not missing anything. A food for thought -- I would like the authors to challenge themselves to come up with a schematic that explains dynamo theory instead of the texts and the scaling laws involved. 

# The magnetic reconnection section is not well explained. Even for review papers , there needs to be some explanation -- especially this becomes important for someone without a MHD background. I would request the authors to provide simple physical explanations/insights as to why magnetic reconnection occurs. 

Some minor points to note:

# Figure 8: does not what the authors are plotting. 

#page 12: Put reference as "In magnetically mediated star formation theory (see [68]-[74]) ...". It is not required to provide the whole sequence of references in a line if they are chronologically arranged 

Author Response

Thank you very much for your comments. We made changes. Fig. 8 represents an evolution of the disk with and without reconnection diffusion. We added text describing this figure and scheme to the dynamo section. We have redone the part about magnetic reconnection and added a description of table 1 about scales and power law into the text of the paper instead of infographics. If it is still needed, we can make it. We made corrections regarding the references.

Round 2

Reviewer 2 Report

I am happy to recommend this manuscript for publication. 

Author Response

Thank you very much for the approval.